# Coenzyme-protein interactions since early life

**Alma Carolina Sanchez Rocha[1], Mikhail Makarov[1], Lukáš Pravda[2†], Marian Novotný[1]\*, Klára Hlouchová[1,3]\***

[1]Department of Cell Biology, Charles University, Prague, Czech Republic; [2]Protein Data Bank in Europe, European Molecular Biology Laboratory, European Bioinformatics Institute (EMBL-EBI), Wellcome Genome Campus, Cambridge, United Kingdom; [3]Institute of Organic Chemistry and Biochemistry, Czech Academy of Sciences, Prague, Czech Republic

**eLife Assessment**

This study presents a **valuable** examination of the prevalence of interactions between amino acids from different periods of Earth's history and coenzymes. While the premise of this work is well founded and the analysis is **solid**, with more data, the interpretation could change. This manuscript would be of interest to evolutionary biologists and biophysicists.

**\*For correspondence:**
marian.novotny@natur.cuni.cz
(MN);
klara.hlouchova@natur.cuni.cz
(KH)

**Present address:** [†]Exscientia, Oxford, United Kingdom

**Competing interest:** The authors declare that no competing interests exist.

**Abstract** Recent findings in protein evolution and peptide prebiotic plausibility have been setting the stage for reconsidering the role of peptides in the early stages of life's origin. Ancient protein families have been found to share common themes and proteins reduced in composition to prebiotically plausible amino acids have been reported capable of structure formation and key functions, such as binding to RNA. While this may suggest peptide relevance in early life, their functional repertoire, when composed of a limited number of early residues (missing some of the most sophisticated functional groups of today's alphabet) has been debated. Cofactors enrich the functional scope of about half of extant enzymes, but whether they could also bind to peptides lacking the evolutionary late amino acids remains speculative. The aim of this study was to resolve the early peptide propensity to bind organic cofactors by analysis of protein-coenzyme interactions across the Protein Data Bank (PDB). We find that the prebiotically plausible amino acids are more abundant in the binding sites of the most ancient coenzymes and that such interactions rely more frequently on the involvement of the protein backbone atoms and metal ion cofactors. Moreover, we have identified a few select examples in today's enzymes where coenzyme binding is supported solely by prebiotically available amino acids. These results imply the plausibility of a coenzyme-peptide functional collaboration preceding the establishment of the Central Dogma and full protein alphabet evolution.

## Introduction

Organic and inorganic cofactors occupy about half of all known protein structures, expanding across all the enzyme E.C. classes (*Putignano et al., 2018*; *Mukhopadhyay et al., 2019*). While their role in current life is indisputable, some of the cofactors were present and apparently crucial also during life's early evolution (*Chu and Zhang, 2020*; *Goldman and Kacar, 2021*; *Kirschning, 2021*; *Fried et al., 2022*; *Kirschning, 2022*). The significance of metal ions has been broadly discussed, regardless of the different origins-of-life scenarios, and has somewhat overshadowed that of organic cofactors (e.g. *Wächtershäuser, 1992*; *Russell and Hall, 1997*; *Lane and Martin, 2012*; *Chu and Zhang, 2020*; *Fried et al., 2022*).

Diverse lines of evidence have, however, indicated that many of the extant organic cofactors (coenzymes) date back to the earliest life, while their core chemistries have been detected in abiotic material, such as recently reported by the Hayabusa2 mission (*Holliday et al., 2007*; *Fried et al., 2022*; *Naraoka et al., 2023*). At the same time, these ancient coenzymes – often of nucleotide origin– have been traced to the most ancient protein folds (such as P-loop NTPases, TIM beta/alpha-barrels, OB, and Rossmann folds) that date before the Last Universal Common Ancestor (LUCA) (*Goldman and Kacar, 2021*; *Caetano-Anollés et al., 2007*; *Goldman et al., 2013*; *Longo et al., 2020a*; *Kessel and Ben-Tal, 2022*). Within the most ancient folds, tens of peptide fragments/themes have been identified throughout seemingly unrelated structural domains and frequently found to mediate ligand binding (*Söding and Lupas, 2003*; *Alva et al., 2015*; *Laurino et al., 2016*; *Kolodny et al., 2021*; *Nepomnyachiy et al., 2017*; *Qiu et al., 2022*). Such themes may well represent the remnants of protoenzymes in a peptide-nucleotide world (*Fried et al., 2022*). *Chu and Zhang, 2020* recently proposed that cofactors could initially 'select' the earliest primitive proteins from the vast sequence space by the ability to bind them. More generally, binding of cofactors to peptides could thus determine the evolution of both protoenzyme function and folding preferences (*Tokuriki and Tawfik, 2009*).

Prior to the fixation of the Central Dogma and ribosomal synthesis, peptides would condense from amino acids (or their alternatives) prebiotically abundant in the environment (*Frenkel-Pinter et al., 2019*; *Frenkel-Pinter et al., 2020*; *Fried et al., 2022*). Independent meta-analyses of the amino acid alphabet evolution based on different possible sources of organic material and different disciplines point towards an 'early alphabet' of ~10 residues (Ala, Asp, Glu, Gly, Ile, Leu, Pro, Ser, Thr, and Val) (*Higgs and Pudritz, 2009*; *Trifonov, 2000*; *Cleaves, 2010*). These could be supplemented by other prebiotically plausible non-canonical amino acids, while the other half of the canonical alphabet is assumed to be the product of later biosynthesis (*Wong and Bronskill, 1979*; *Weber and Miller, 1981*; *Burton et al., 2012*; *Zaia et al., 2008*). Typically, the early amino acids (canonical as well as non-canonical) are smaller and less complex, missing e.g., sulfur groups and aromatics. Additionally, the canonical early alphabet lacks positively charged residues. An emerging question, therefore, is whether coenzymes could bind to small proteins of prebiotic relevance and whether they could be bound by the prebiotically available residues. In such a scenario, cofactors would provide a palette of functional groups to the early peptide world, which would nominate them relatively sophisticated structural and catalytic hubs (*Milner-White and Russell, 2011*). Alternatively, if coenzymes could not

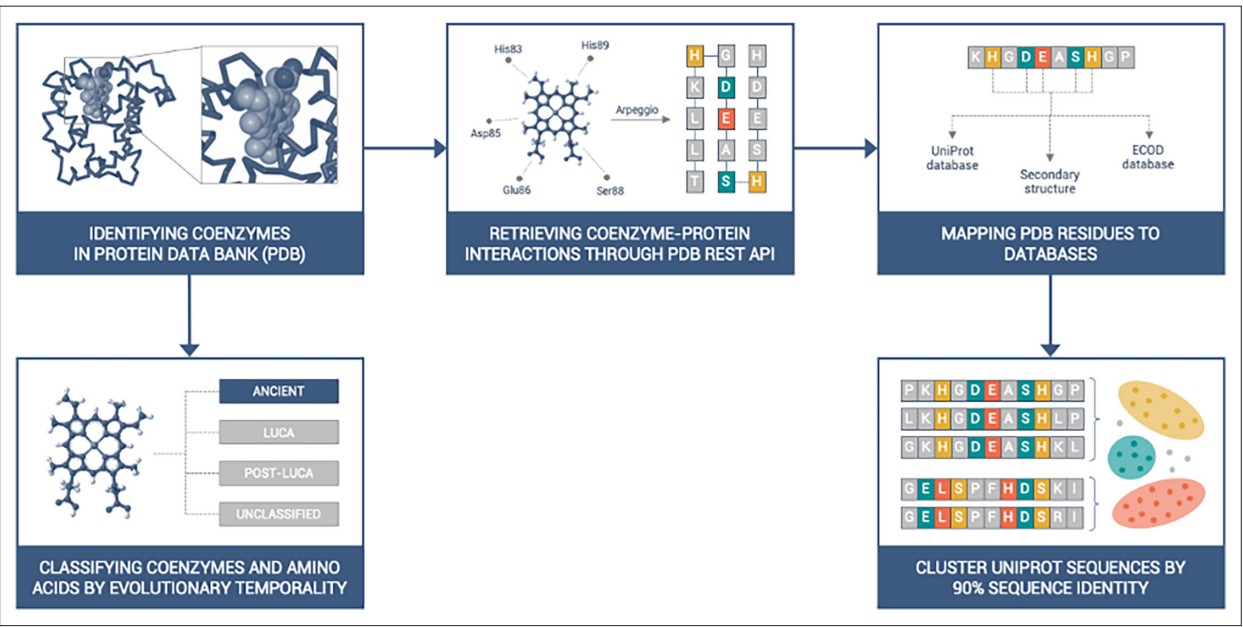

**Figure 1.** Workflow of this study. All available coenzymes in the Protein Data Bank (PDB) were identified according to the CoFactor database (*Fischer et al., 2010*). The PDB entries of structures bound to coenzymes were downloaded programmatically through the PDBe REST API (pdbe.org/api), including the interatomic cofactor-protein interactions, calculated by Arpeggio (*Jubb et al., 2017*). The coenzyme-binding amino acids were mapped to UniProt databases via Structure Integration with Function (SIFTS) (*Velankar et al., 2013*; *Dana et al., 2019*). PDB entries were grouped by UniProt code; redundancy was removed by clustering the UniProt sequences by 90% (and in parallel also 30%) sequence identity.

be bound by these simple amino acids, this would suggest that their pairing with peptide molecules would become relevant only after the evolution of the full amino acid alphabet.

Work from our group and others has recently demonstrated that in select cases, protein sequences re-engineered from the early amino acids can still bind to nucleic acid and nucleotide-based cofactors (*Longo et al., 2020b*; *Makarov et al., 2021*; *Giacobelli et al., 2022*). Whether this phenomenon is still seen in today's biology, its abundance and laws represent open questions. Here, we present a systematic survey of coenzyme binding throughout the PDB database. The outcomes of our study support that the coenzyme binding characteristics of amino acids differ by their evolutionary age. Early amino acids are enriched in binding pockets of the most ancient coenzymes and the interaction relies predominantly on the protein backbone groups. Selected examples show that, unlike evolutionary younger cofactors, the ancient cofactors can still be bound in proteins only by early amino acids. Our analysis, therefore, points to an early peptide-coenzyme significance, preceding the evolution of proteosynthesis and the fixation of the Central Dogma.

## Results
### Identification of coenzymes in PDB
We identified all the available structures from the PDB that interact with the 27 coenzyme classes as defined in *Fischer et al., 2010*. In addition, ATP (that was not included in that study) was included here. Using these parameters, we found 25,822 protein structures and 81 nucleic acid macromolecules (*Supplementary file 1*, *Supplementary file 2*). The protein structures were assigned to 8194 UniProt (*Bateman et al., 2023*) codes. Those UniProt sequences were clustered by 90% identity and resulted in 7399 unique UniProt entries, corresponding to 21,317 protein structures (*Figure 1*). In parallel, the clustering was also performed for 30% sequence identity, resulting in 3544 UniProt codes and 9645 PDB structures.

The interaction ratio method was adopted to identify the most relevant residues in coenzyme binding sites. For each protein (unique UniProt ID), we defined the cofactor binding site as a subset of amino acids that appeared to interact with the cofactor in at least 50% of the structures of that particular protein within our dataset to pinpoint the amino acids that are important for the interaction. This methodology does not consider any qualitative criteria (e.g. resolution, R-factor, Clashscore).

Our database is composed of protein structures from members of the three cellular domains – Bacteria (54.3%), Archaea (6.2%), Eukaryota (37.8%) – as well as Viruses (1.5%), metagenomes, and unassigned entries (0.3%) (*Supplementary file 2*).

### Evolutionary classification of coenzymes
To differentiate the evolutionary age of the analyzed coenzymes, we further adapted the classification system from *Fried et al., 2022*. This system encompasses four primary categories (or temporalities) and one additional subcategory: (i) 'Ancient' coenzymes, including the subcategory 'Nucleotide derived;' (ii) 'LUCA' coenzymes; (iii) 'Post-LUCA' coenzymes; and (iv) 'Unclassified' coenzymes (*Figure 2*).

'Ancient' coenzymes comprise those that could be prebiotically synthesized, according to available studies (*Miller and Schlesinger, 1993*; *Keefe et al., 1995*; *Holliday et al., 2007*; *Kirschning, 2021*; *Menor-Salván et al., 2022*; *Pinna et al., 2022*); while the subcategory 'Nucleotide derived' includes cofactors chemically derived from nucleotides (*White, 1976*; *Monteverde et al., 2017*). 'LUCA' coenzymes were presumably present in the last universal common ancestor (LUCA) and exhibit a universal distribution among Bacteria, Archaea, and Eukarya, although their prebiotically feasible synthesis was not established. 'Post-LUCA' coenzymes likely originated only after the divergence of the three cellular domains, mirrored in their non-universal distribution. 'Unclassified' coenzymes do not conform to the classification scheme. As a typical representative of the latest category, Coenzyme M has been synthesized under prebiotic conditions (*Miller and Schlesinger, 1993*; *Kirschning, 2021*); nonetheless, its biosynthetic pathways in Archaea and Bacteria have been shown to arise through convergent evolution, and it is mainly prevalent in methanogens (*Wu et al., 2022*). Factor F430 is a coenzyme only distributed in methanogens (*Thauer and Bonacker, 1994*), although its precursors have been synthesized prebiotically (*Seitz et al., 2021*). Glutathione is another example of a coenzyme with restricted biological distribution, being mainly in eukaryotes, Gram-negative bacteria, and one archaea phylum (*Copley and Dhillon, 2002*), and the feasibility of its prebiotic synthesis remains unclear (*Bonfio et al.,*

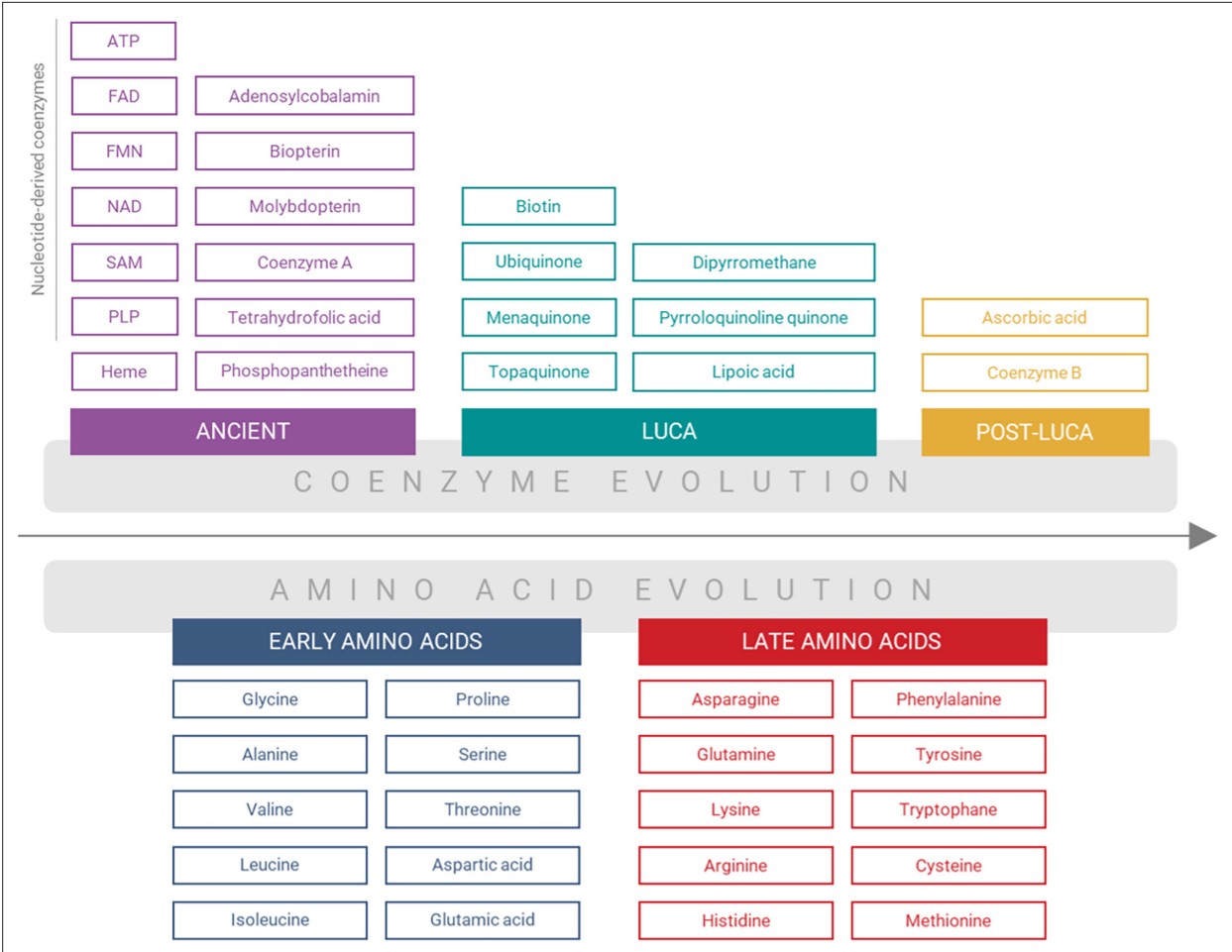

**Figure 2.** Classification of coenzymes and amino acids by their assumed evolutionary temporality. The 'Unclassified' coenzymes Thiamine diphosphate, Coenzyme M, Factor F430, and Glutathione are not shown in the scheme.

The online version of this article includes the following figure supplement(s) for figure 2:

**Figure supplement 1.** Enzymatic class diversity per coenzyme class.

*2017*). Thiamine diphosphate was also designated as Unclassified. Although the definitive prebiotic synthesis of thiamine diphosphate remains unclear, preliminary investigations conducted by *Aylward, 2006a* and *Aylward and Bofinger, 2006b* suggest its presence in the prebiotic world. Its nucleotide nature (*White, 1976*) and the existence of its universal riboswitch (*Barrick and Breaker, 2007*) provide compelling evidence of its potential status as an Ancient coenzyme.

The ancient coenzymes represent the most abundant temporality of our PDB dataset, dominated by ATP, NAD, Heme, FAD, SAM, and CoA structures and amounting to 94% of all analyzed structures in our database grouped by UniProt codes. Within the enzyme E.C. classification, oxidoreductases and transferases represent the classes with the most abundant coenzyme content. While the LUCA, Post-LUCA, and Unclassified coenzymes are typically found in specific enzyme classes, the Ancient coenzymes are distributed across all the E.C. classes (*Figure 2—figure supplement 1*).

## Distribution of amino acids in the coenzyme binding sites

We hypothesized that the evolutionary significance of individual coenzyme classes would be reflected in distinct amino acid binding propensities, as a smaller 'early' protein alphabet apparently preceded its canonical version. The abundance of residues that compose each coenzyme binding site was analyzed and examined with respect to the order by which individual amino acids have been reported to enter the protein canonical alphabet (*Higgs and Pudritz, 2009*; *Figure 3*; *Supplementary file 3*). The binding site composition for both 90% and 30% identity datasets revealed that the occupancy

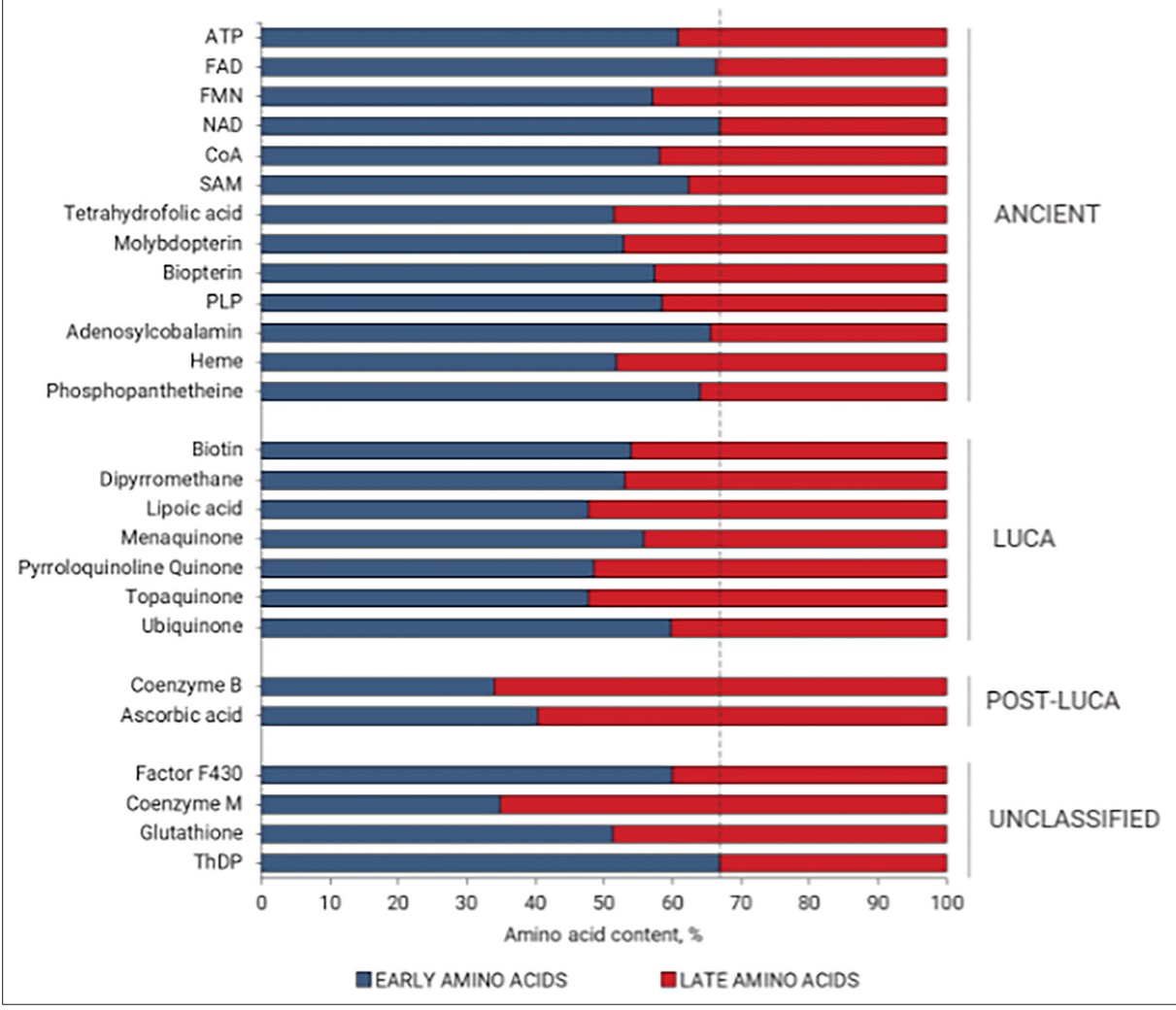

**Figure 3.** Early versus late amino acid composition of the coenzyme binding sites, categorized according to the evolutionary temporality of coenzymes. Early amino acids are shown in color blue and late residues in red. The dashed line corresponds to the proportion of early vs. late amino acids within the UniProt composition of the sequences derived from our database (67% early and 33% late residues). The statistical significance of the early versus late amino acid composition was assessed by a Chi-squared test (*p*<0.0001). Detailed statistical data are listed in ***Supplementary file 9***.

of early amino acids is higher in Ancient coenzyme binding sites and tends to decrease in LUCA and Post-LUCA cofactor sites. Overall, for the 90% dataset, the average occupancy of early vs. late amino acids in the Ancient sites is 61% vs 39% while this ratio decreases to 53% vs 47% for the LUCA and 47% vs 53% in Post-LUCA sites. These numbers follow the same trend for the 30% identity dataset and throughout the rest of the analysis, the 90% identity dataset – which includes a higher number of proteins – was evaluated for more robust statistical analysis.

To explore the contribution of individual amino acids to this effect, fractional difference (FD) for early vs. late amino acids among the Ancient, LUCA, and Post-LUCA coenzyme binding was calculated (***Supplementary file 6A***). The mean FD revealed a similar trend to the amino acid composition analysis (***Figure 3***). The amino acids most enriched in LUCA vs. Post-LUCA are Gly, Ser, and Leu (FD of 4.4, 4.3, and 4.1, respectively), while the most depleted include Phe, Arg, and His (FD of −11,−4.2, and −3.2) (***Supplementary file 6B***).

Moreover, we investigated whether the observed trend in amino acid occurrence at the binding sites was dominated by the presence of phosphate groups, which are common in many ancient cofactors except for SAM, tetrahydrofolic acid, biopterin, and heme. An additional analysis, therefore, excluded all phosphate-containing coenzymes, indicating that while the trend is less pronounced, it remains even in the absence of phosphate groups (***Supplementary file 7***).

To examine the impact that the distribution of amino acids in the coenzyme binding sites has on the binding modes, the interactions between coenzymes and individual amino acids were further inspected.

## Interaction types between coenzymes and proteins

First, backbone vs side chain protein interactions of all the coenzyme classes were mapped (*Figure 4A*). As expected, most of the interactions with coenzymes are mediated by amino acid side chains (61%), frequently in combination with backbone (24%) (*Figure 4*). Nevertheless, purely backbone interactions prevail in Ancient coenzymes (24%) (*Figure 4A*). When backbone interactions are present throughout the different coenzyme temporalities, they are dominated by the early amino acids (*Figure 4B*).

Next, we inspected the interaction types for each amino acid-coenzyme binding event employing Arpeggio (*Jubb et al., 2017*). The analysis revealed that electrostatic interactions are dominant in all coenzyme ages (*Figure 4—figure supplement 1*). In Ancient cofactors, electrostatic interactions are more frequently mediated by early residues. This trend is more significant for the nucleotide-derived Ancient coenzymes. The second most prevalent interaction type is Van der Waals for Ancient cofactors, while hydrophobic interactions are similarly frequent as Van der Waals in the LUCA and Post-LUCA classes.

## Structural properties of cofactor binding sites

Structural properties of proteins have been observed to change during eons of life's evolution (*Edwards et al., 2013*; *Lupas and Alva, 2017*; *Kovacs et al., 2017*). To map its interdependence with the binding of cofactors, the secondary structure and fold classes of individual coenzyme binding sites were analyzed here.

There are detectable differences in the binding site secondary structure content among the coenzyme temporalities (*Figure 5*, *Figure 5—figure supplement 1*). While loops and helices dominate all the binding sites, they are less represented in the Ancient and LUCA coenzyme sites which are more rich in beta-sheet structures (*Figure 5*). This distinction is found only on the level of the binding sites and not preserved on the level of the overall protein structure.

To explore the fold diversity of domains containing the coenzyme binding sites, we assigned their ECOD X-groups (*Cheng et al., 2014*) at a residue level (*Supplementary file 4*). In total, 101 groups were identified. The ancient coenzymes are associated with higher numbers of different X-groups than the LUCA and Post-LUCA cofactors (*Figure 6*). Some coenzymes stand out by their large number of associated folds: the Ancient ATP (74); CoA (34); NAD (30); Heme (27), and the Unclassified cofactor Glutathione (25).

The most frequently observed X-groups include Rossmann-like, Alpha-beta plaits, TIM beta/alpha-barrel, Flavodoxin-like, cradle loop barrel, HUP domain-like, and beta-Grasp (*Figure 6*). Among these, Rossmann-like, TIM beta/alpha-barrel, and Flavodoxin-like bind to coenzymes of all ages.

## Coenzyme early vs late binding sites

To further explore whether extant proteins can bind enzymes only by early or only by late residues (featuring early vs late binding sites), we looked for these specific cases and analyzed their evolutionary conservation.

We found 25 PDB entries that contain at least one chain bound to coenzymes solely by early amino acids (*Figure 7*; *Figure 7—figure supplement 1*). Those structures correspond to 17 different proteins, represented by unique UniProt codes. The full set of those proteins binds exclusively to ancient coenzymes: ATP, NAD, and phosphopantetheine. In comparison, 15 PDB entries, representing 12 unique proteins, bind coenzymes only by late amino acids (*Figure 7*; *Figure 7—figure supplement 1*). These examples include all Ancient-to-Post-LUCA coenzymes: ATP, CoA, NAD, PLP, biotin, and ascorbic acid.

To assess the conservation of amino acids in these specific binding sites, we used ConSurf (*Ashkenazy et al., 2010*; *Ashkenazy et al., 2016*). According to the analysis, both the early and late binding sites are relatively highly conserved. Around 60% of the residues from both cases have conservation scores ≥7. Furthermore, we employed the *MAX AA* parameter, that represents the most abundant residue in the multiple sequence alignment of all homologs. 76 vs 72% of the residues in the early vs late binding sites are the same, which suggests their evolutionary conservation in both cases.

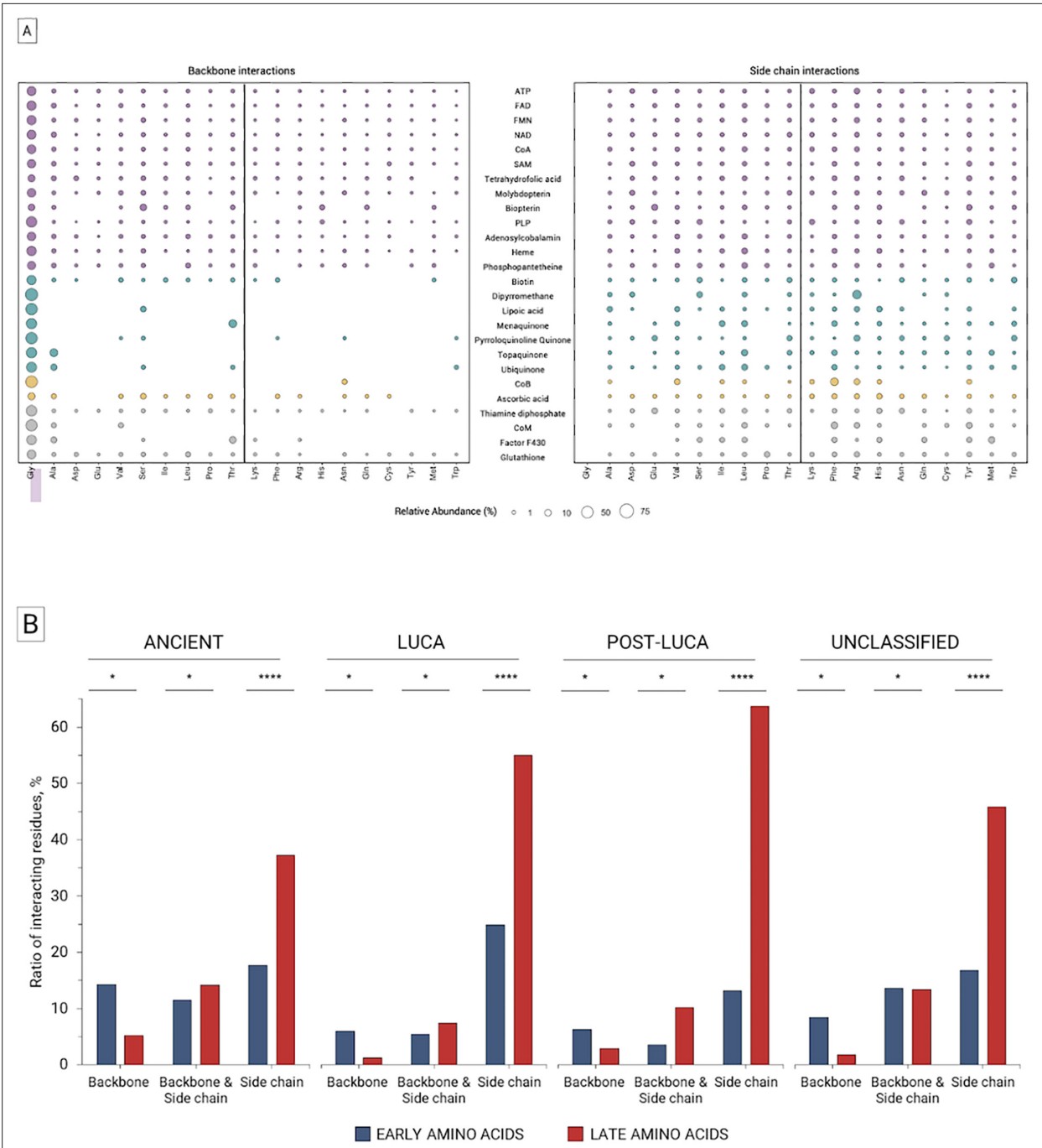

**Figure 4.** Binding of coenzymes with early and late amino acids by backbone and side chain atoms. 'Backbone' interactions refer to residues in the coenzyme binding sites that interact purely through amino acid backbone atoms. 'Side chain' interactions involve residues that interact solely via side chain atoms. 'Backbone & Side chain' residues are those that interact with the coenzyme using both their backbone and side chain atoms. (**A**) Abundance of amino acids in individual studied coenzymes. 'Backbone & Side chain' interactions are not depicted. Unclassified cofactors are in gray, Post-Last Universal Common Ancestor (LUCA) in yellow, LUCA in cyan, and Ancient in purple. Amino acids are ranked by the order of addition of amino acids to the genetic code (**Higgs and Pudritz, 2009**). (**B**) Proportion of early versus late residues in coenzyme categories by interaction type. In each coenzyme category, the individual proportions add up to 100%. The amino acid composition was normalized by the percentage of late residues from the UniProt sequences retrieved from our database. The statistical significance of early versus late amino acid composition for each interaction type per coenzyme temporality was determined by a Chi-squared test (*$p<0.05$; **$p<0.01$; ***$p<0.001$; ****$p<0.0001$). For detailed statistical analysis, refer to **Supplementary file 10**.

The online version of this article includes the following figure supplement(s) for figure 4:

**Figure supplement 1.** Interaction types.

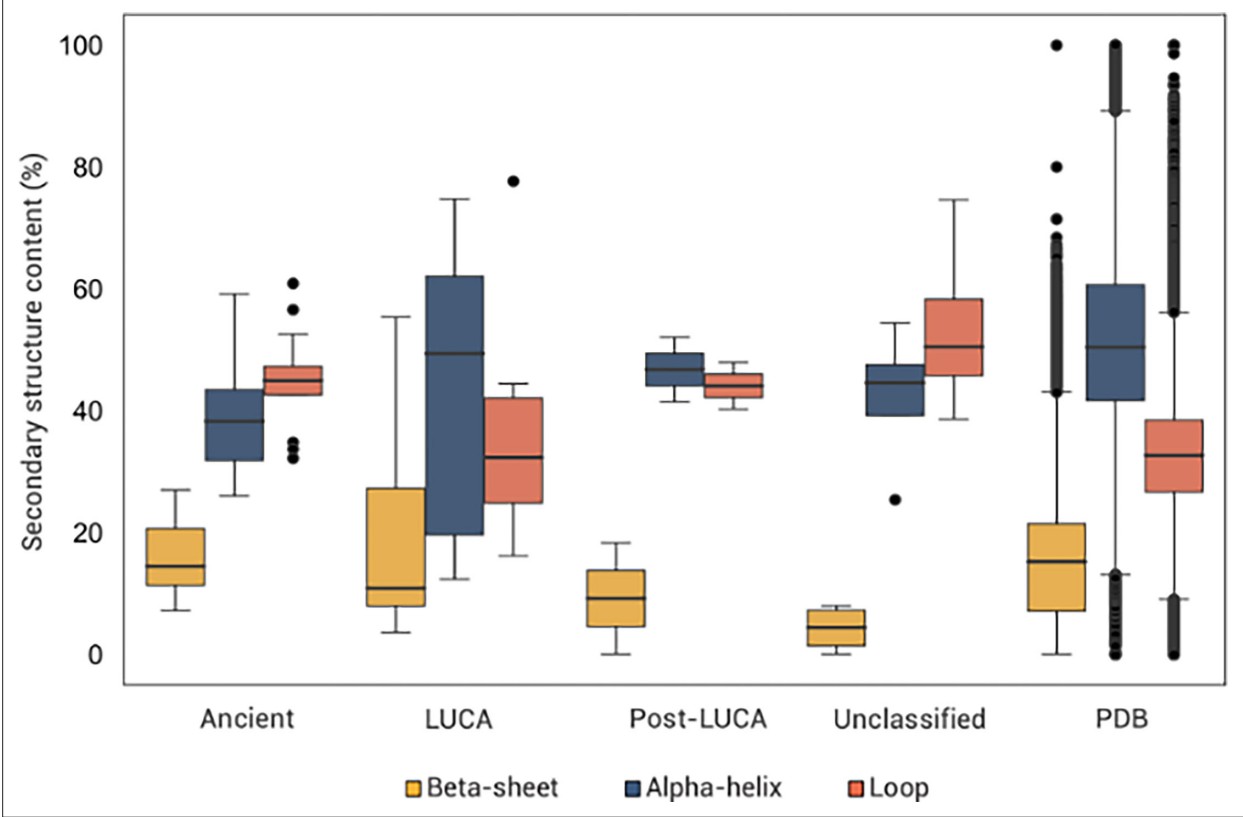

**Figure 5.** Secondary structure content in coenzyme binding sites. Composition of secondary structural elements in amino acids interacting with coenzymes. The Protein Data Bank (PDB) category represents secondary structure content across the dataset for comparison with coenzyme binding sites. Additional statistical analyses are shown in *Supplementary file 11*.

The online version of this article includes the following figure supplement(s) for figure 5:

**Figure supplement 1.** Secondary structure content per cofactor class.

### Coenzyme binding mediated by metal ions

Because of the significance of metal ions in both extant and early life, we also analyzed coenzyme binding via metal ions (*Supplementary file 5*). Notably, this phenomenon is more frequent in Ancient cofactors, constituting approximately 24% of coenzyme binding sites that have at least one metal ion. Younger cofactors exhibit a lower requirement for metal ion binding. LUCA coenzymes exhibit a metal ion binding in approximately 13% of instances, while in Post-LUCA, 11% and in Unclassified coenzymes, about 23%. Certain coenzymes exhibit a notably high percentage of cases reliant on at least one $Mg^{2+}$ ion (76% in the case of Thiamine diphosphate and 55% in the case of ATP binding). The subsequent most prevalent mediating ion is $Ca^{2+}$, found along with 65 % cases of the LUCA coenzyme Pyrroloquinoline Quinone. Following $Ca^{2+}$, the next most frequent metal ions mediating coenzyme binding are $Mn^{2+}$ and $Fe^{2+}$.

### Discussion

Enzymatic activities rely heavily on interplay with organic cofactors. Those are found at the very heart of cellular metabolism, and some of them quite possibly branch deep to life's early start (*White, 1976*). The core chemistries of the most abundant and ancient coenzymes have been repeatedly detected in material and experiments mimicking prebiotic environments (*Miller and Schlesinger, 1993*; *Keefe et al., 1995*; *Holliday et al., 2007*; *Kirschning, 2021*; *Menor-Salván et al., 2022*; *Pinna et al., 2022*; *Fairchild et al., 2024*). Along with metal ions and minerals, some of the extant coenzymes could probably catalyze metabolic reactions in the absence of enzymes, before their emergence (*Muchowska et al., 2020*; *Henriques Pereira et al., 2022*; *Cvjetan et al., 2023*; *Dherbassy et al., 2023*). When

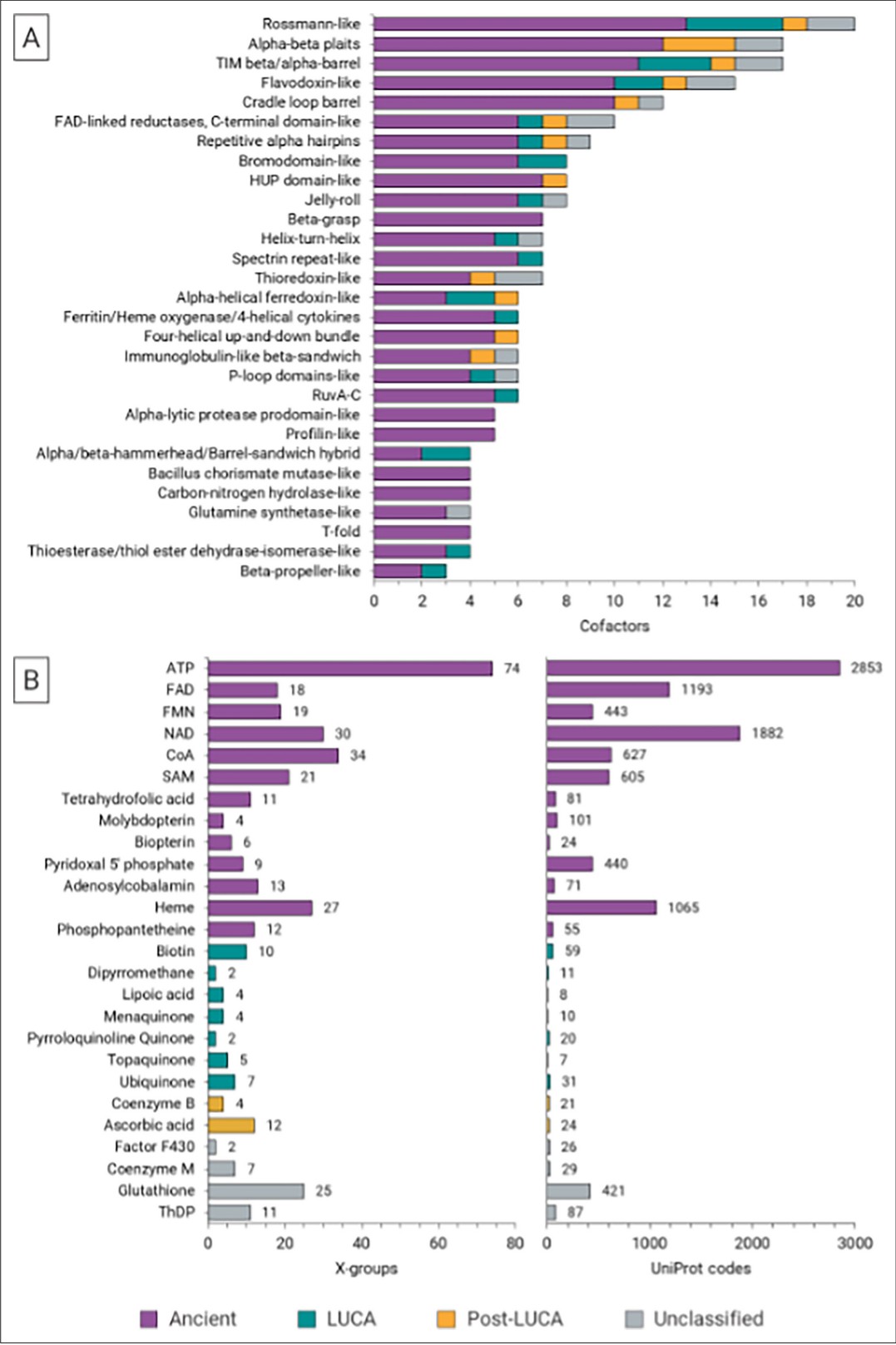

**Figure 6.** Fold diversity of coenzyme binding sites. (**A**) Folds represented by ECOD X-groups, according to numbers of coenzyme binding sites. (**B**) Comparison of numbers of ECOD X-groups vs. UniProt entries per cofactor class.

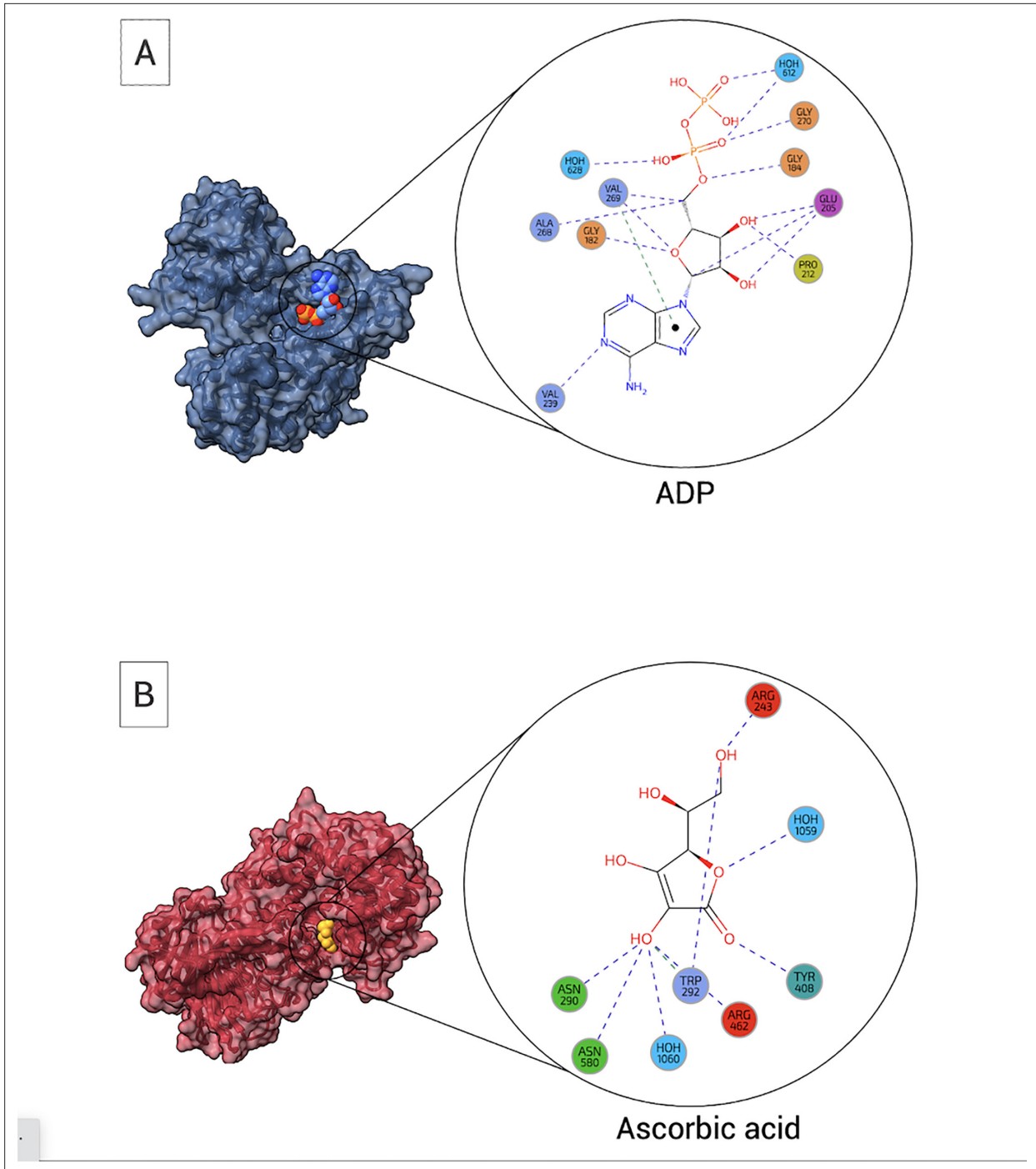

**Figure 7.** Examples of coenzyme binding solely through early or late amino acids. (**A**) Coenzymes bound exclusively by early residues AMP bound by ATP-phosphoribosyltransferase. PDB code 6czm (chain B) created by LIGPLOT (*Laskowski and Swindells, 2011*). (**B**) Coenzyme, entirely bound by late residues (Ascorbic acid bound by Hyaluronate lyase. Protein Data Bank (PDB) code 1f9g (chain A), created by LIGPLOT).

The online version of this article includes the following figure supplement(s) for figure 7:

**Figure supplement 1.** Cofactor binding with only early amino acids.

and how coenzymes seeded the functional hubs of today's enzymes represent a fundamental bridge between prebiotic chemistry and biochemistry and, therefore, one of the central questions in the study of life's origins (*Preiner et al., 2020*).

The aim of this study was to resolve this conundrum by analyzing protein-coenzyme interactions throughout PDB with respect to amino acid and coenzyme evolutionary temporality (*Figure 2*). While no direct inferences about the distant evolutionary past can be drawn from the analysis of extant proteins, the principles guiding these interactions can imply their potential prebiotic feasibility and significance.

## Ancient coenzymes are abundant in extant life and bind more frequently to early amino acids

We find that an absolute majority (94%) of extant coenzymes that appear in PDB structures is conserved across all the life's domains and available already through prebiotic syntheses (i.e. Ancient). This coenzyme temporality bears many molecules that are derived from nucleotides (such as ATP, FAD, SAM, and NAD) and is spread throughout all the E.C. classes of protein catalysts. This is the first obvious distinction from the other (less-populated) classes of coenzymes and supports the coenzyme-peptide significance from the earliest life until today (*White, 1976*; *Narunsky et al., 2020*).

There are several outstanding differences in the properties of binding to proteins among the different coenzyme temporalities. First, the ancient coenzymes bind to proteins more frequently via early amino acids. On average, this interaction presents 61% of all Ancient coenzyme bonds to proteins. It is only 53 and 38% for the LUCA and Post-LUCA coenzymes, respectively.

While the Ancient nucleotide-derived and Unclassified coenzyme binding sites are dominated by residues in the loop conformation (*Figure 5*; *Figure 5—figure supplement 1*), there is also a substantially higher frequency of residues in beta-sheet conformations when compared to Post-LUCA coenzyme sites. Those, on the other hand, are dominated by alpha-helical conformations. Loops exhibit greater sequence variability compared to ordered structures, and their flexible nature enables them to undergo structural changes (*Kessel and Ben-Tal, 2018*; *Tokuriki and Tawfik, 2009*). Such properties of protein active sites were previously associated with evolvability and promiscuity (*Tokuriki and Tawfik, 2009*; *Corbella et al., 2023*). The role of loops could thus be important for the flexibility and versatility of early peptide-coenzyme binding sites. It has been noted that evolutionary benefits would be presented by sequences that could adopt closed-loop conformations, providing stability and protection to early coenzymes, and such hubs could truly transition the peptide-coenzyme world towards primordial enzymes (*Goncearenco and Berezovsky, 2010*; *Gamiz-Arco et al., 2021*; *Toledo-Patiño et al., 2022*; *Gutierrez-Rus et al., 2023*). Besides sequences without regular secondary structure elements, beta-sheets have been considered a more prevalent and significant motif during early stages of protein evolution than alpha helices. Beta sheet represents the first structural motif in models of the ribosomal evolution, and it has also been observed as a mildly enriched motif in sequences formed from early amino acids (*Brack and Orgel, 1975*; *Lupas and Alva, 2017*; *Kovacs et al., 2017*; *Tretyachenko et al., 2022*). Ancient coenzyme-peptide binding properties support the scenario of its significance during early stages of protein evolution.

## Ancient coenzymes bind to proteins through more backbone interactions, typically assigned to early amino acids

While all the coenzymes bind preferentially to protein residue side chains, more backbone interactions appear in the Ancient coenzyme temporality when compared to others. This supports an earlier hypothesis that functions of the earliest peptides (possibly of variable compositions and lengths) would be performed with the assistance of the main chain atoms rather than their side chains (*Milner-White and Russell, 2011*). A specific example of such a scenario was recently reported, where a dihydrofolate reductase activity was supported purely by protein backbone-coenzyme interactions (*Lemay-St-Denis and Pelletier, 2023*). Finally, Longo et al. recently analyzed binding sites of different phosphate-containing ligands, which were arguably of high relevance during the earliest stages of life, connecting all of today's core metabolism (*Longo et al., 2020b*). They observed that, unlike the evolutionary younger binding motifs (which rely on side-chain binding), the most ancient lineages indeed bind to phosphate moieties predominantly via the protein backbone.

Our analysis assigns this phenomenon primarily to interactions via early amino acids that (as mentioned above) are generally enriched in the binding interface of the Ancient coenzymes. This implies that late amino acids would not necessarily be needed for the sovereignty of coenzyme-peptide interplay. To address this intriguing possibility, we next searched whether there are such examples in the PDB dataset, where coenzymes would be bound exclusively by early amino acids. We found 17 such proteins where all the coenzymes belonged to the Ancient temporality (such as ATP and NAD). Together with all of the above, this finding supports the possibility that peptide-coenzyme functional hubs could have originated before the evolution of the full canonical amino acid alphabet.

Reinforcing this, we have recently demonstrated on a select example of a ribosomal RNA-binding domain that a negatively charged variant of the protein composed of only early amino acids is indeed capable of binding to RNA (*Giacobelli et al., 2022*). In that case, the interaction is further supported by metal ions that were not present at the binding interface of the wild-type protein. Interestingly, the same trend was observed here throughout the PDB dataset. 24% of the Ancient coenzymes in PDB are additionally mediated by at least one metal ion. LUCA and Post-LUCA coenzymes involved metal ions only in 13 and 11%, respectively. It is quite probable that not all the metal ion densities are recognized or fully resolved in all the PDB structures that were used in our analysis. Nevertheless, we hypothesize that the overall trend can be attributed to the inherent negative charge of many of the Ancient coenzymes, necessitating engagement of positively charged metal ions. This would also be true for direct interaction of early peptides/proteins and metal ions, independent of organic cofactor involvement, as discussed previously by us and others (*Bromberg et al., 2022*; *Frenkel-Pinter et al., 2020*; *Fried et al., 2022*). For example, it has been observed that coordination of prebiotically most relevant metal ions (e.g. $Mg^{2+}$) is more often mediated by early amino acids such as Asp and Glu, whereas metal ions of later relevance (e.g. $Cu^+$ and $Zn^{2+}$) bind more frequently via late amino acids like His and Cys (*Fried et al., 2022*). Similarly, ancient metal binding folds have been shown to be enriched in early amino acids (*Bromberg et al., 2022*). Along with the general adaptive properties of late amino acids in expanding the chemistry space, the late amino acids would supplement the positive charge in the residue side chains (*Ilardo and Freeland, 2014*).

## Coenzymes could have served as bridges towards protein structural and functional sovereignty from the peptide-nucleotide world

Our study further revealed that ancient coenzymes stand out in the variety of protein structures that they bind to, as represented by the ECOD X-groups. While this may partly be caused by their general over-representation in our dataset, the significance of coenzymes has been pointed out previously throughout the most ancient protein folds, such as P-loop NTPases, TIM beta/alpha-barrels, OB, and Rossmann folds (*Caetano-Anollés et al., 2007*; *Goldman et al., 2013*; *Longo et al., 2020a*; *Longo et al., 2020b*). It has been implied by computer simulations that coenzymes could bind to proteins with similar propensity even before the onset of protein homochirality, despite lower structural stability and secondary structure content in heterochiral polypeptides (*Skolnick et al., 2019*).

Phosphate-containing coenzymes truly stand out by their large number of associated folds (e.g. 74 different ECOD X-groups in the case of ATP). It has been postulated that phosphate-binding loops served as the most significant precursors for contemporary enzymes (*Romero Romero et al., 2018*). Combining ancestral sequence reconstruction and selection/protein design, short polypeptide motifs capable of poly/nucleotide binding have been recovered from the P-loop NTPase and HhH motifs, relying primarily on early amino acids (*Longo et al., 2020b*; *Romero Romero et al., 2018*). Both demonstrated that these ancient motifs are highly robust to sequence variations, implying that such interaction can be encountered more easily than previously thought (*Longo et al., 2020b*; *Keefe and Szostak, 2001*). Provokingly, it has also been implied that many of these motifs emerged initially as polynucleotide binders and started serving catalysis only after gaining higher structural complexity (*Romero Romero et al., 2018*).

Our entry search of PDB coenzymes also retrieved 80 RNA structures. While these were not the primary subject of our analysis, the majority of those were found to interact with Ancient coenzymes, mainly the nucleotide-derived ones, such as ATP, FMN, NAD, and SAM (*Supplementary file 8*). While some of these structures are assigned as riboswitches, other coenzyme-RNA complexes belong to ribozymes, representing the potential of polynucleotides in early catalysis as discussed by many previous studies (*White, 1976*; *White, 1982*; *Gilbert, 1986*; *Reyes-Prieto et al., 2012*; *Goldman*

*and Kacar, 2021*). If peptide-polynucleotide interactions were initially more feasible and dominant (in a putative peptide-polynucleotide world, as implied above), coenzymes could have played a key role in resolving the sovereignty of these molecules towards their tertiary structures and catalytic functions. Polypeptide-coenzyme catalysts would soon dominate in performance (enabling more efficient catalysis) and functional repertoires, especially those that would be hard to facilitate in their absence, such as oxidoreductases and transferases (*Fried et al., 2022*; *Goldman and Kacar, 2021*; *Kessel and Ben-Tal, 2018*).

## Limitations of our work

The first obvious drawback of this analysis is the ambiguity that accompanies the division of coenzymes into evolutionary age temporalities. While the prebiotic availability (or lack of) is quite consensual in some cases, there are also contradictory studies and opinions in the case of some other coenzymes. Some of the coenzymes have, e.g., prebiotic precursors but are not present across all the kingdoms of life. This may suggest that the coenzyme became important only post-LUCA but it can also mean that its importance was only preserved in specific branches of life. 'Unclassified' category has been included for such specific cases (presented e.g. by glutathione). Several properties of glutathione that were identified here (such as protein backbone vs. sidechain interaction, ubiquity across all E.C. enzyme classes and a high number of associated X-groups) would suggest that it is closer to Ancient coenzymes. The other 'classified' coenzymes were categorized based on prevailing studies, although some ambiguity remains. Despite our efforts, the evolutionary ratings of coenzymes (and amino acids) are, therefore, not always clear-cut and not all belong to the categories with the same weight (e.g. some are likely to be more ancient than others). For example, the nucleotide-derived coenzymes probably predate the others of the Ancient temporality - it has been proposed that S-adenosylmethionine emerged before the more complex heme-related porphyrin adenosylcobalamin or coenzyme B12 functionalities (*Lazcano, 2012*).

Another possible bias of our study stems from the population differences among the three coenzyme temporalities. The ancient coenzymes are by far the most abundant temporality in the PDB dataset. It can be argued that this is the result of their ~4 billion years of essentiality to life (*Goldman and Kacar, 2021*). Nevertheless, it may as well be contributed by the bias of structures that are deposited in the PDB and most probably does not reflect the true distribution of coenzymes in the biological protein space. Additionally, the comparison of differentially populated temporalities has challenged some aspects of the analysis presented here, although care has been taken to perform all the appropriate statistical tests.

## Conclusions

The findings presented here propose that early (the less complex and prebiotically plausible) amino acids are sufficient for binding to Ancient coenzymes. Consequently, coenzyme-peptide interactions might have been conceivable at a time when the amino acid alphabet was not yet evolved to its current form. In such interactions, binding modes would rely more on the protein backbone atoms and on involvement of metal ions, both of which are less frequent in interactions with evolutionarily young coenzymes.

## Materials and methods

### Identification of organic cofactors

We systematically identified all available cofactor and cofactor-like molecules in the Protein Data Bank in Europe (wwPDB *Varadi et al., 2020*) programmatically through the PDBe REST API (pdbe.org/api) (*Mukhopadhyay et al., 2019*). All cofactor molecules were classified into 27 classes based on the CoFactor database (*Fischer et al., 2010*). Furthermore, we included ATP and its analogs as an additional cofactor class.

The identification of all the available ligand codes from the PDB chemical component dictionary for each cofactor class was achieved by programmatic access through the 'Cofactors' endpoint (https://www.ebi.ac.uk/pdbe/api/pdb/compound/cofactors) using the PDBe REST API (pdbe.org/api) and all responses were in JSON format.

## Structural database and classifications

We retrieved the PDB entries associated with each chemical component from every cofactor class using the 'PDB entries containing the compound' endpoint (https://www.ebi.ac.uk/pdbe/api/pdb/compound/in_pdb/:id) via the Entry-based API. The count of PDBe entries for each cofactor class is provided in the supplementary information (*Supplementary file 1*). The information from the REST API was unavailable for two coenzyme classes, MIO and Orthoquinone, so they were excluded from the analysis.

The secondary structure assignments and (EC) numbers for all PDB structures analyzed were determined through residue-level cross-references obtained from the SIFTS XML files (*Velankar et al., 2013*; *Dana et al., 2019*). Secondary structure elements include 'h' for helix, 'b' for strand, and 'c' for coil; and they correspond to the information available in the PDBe website. Only observed residues were examined.

Furthermore, we assigned all our PDB entries to the ECOD hierarchical system groups 'X,' 'H,' and 'F' (*Cheng et al., 2014*).

## UniProt assignment and interaction ratio

The assignment of UniProt codes to our structural dataset was achieved by mapping the information with the SIFTS (*Velankar et al., 2013*; *Dana et al., 2019*) file 'pdb_chain_uniprot.tsv.gz' (https://www.ebi.ac.uk/pdbe/docs/sifts/quick.html). Next, we mapped the UniProt residue to each of the PDB structures with the residue-level cross-reference data of SIFTS by retrieving the XML files (https://www.ebi.ac.uk/pdbe/docs/sifts/quick.html).

Each UniProt code represents a unique protein sequence that encompasses one or various PDBe-associated entries. With the aim of filtering those residues relevant to the interaction sites at the level of protein sequence, we incorporated the interaction ratio. The interaction ratio is a measure of the interaction for each ligand with all its PDB-associated entries by UniProt residue. Those residue-ligand interactions that were preserved in more than 50% of the associated PDB entries were selected (we call the ratio of preserved interactions among structures of one unique protein interaction ratio).

Upon UniProt residues assignment for each residue of the PDB structures, we downloaded the calculated interaction ratio with the endpoint 'UniProt- Get ligand binding residues for a UniProt accession' .

The redundancy of our database was removed by clustering the UniProt sequences using CD-HIT (*Li and Godzik, 2006*) with a 90% sequence identity parameter.

## Analysis of coenzyme interactions

In order to analyze the amino acid-coenzyme interactions, we used the PDBe Aggregated API (https://www.ebi.ac.uk/pdbe/graph-api/pdbe_doc/). For each PDB entry, we first retrieved all bound molecules via the "PDB – Get bound molecule interactions" endpoint of the Aggregated API. We then obtained the ligand interactions for each bound molecule using the "PDB – Get bound ligand interactions" endpoint, also provided by the same Aggregated API, which calculates these interactions with Arpeggio (*Jubb et al., 2017*). The retrieved interactions included the standard amino acid codes, water molecules, and metal ions.

We classified all the interactions reported by Arpeggio into nine distinct interaction types. The classification scheme aligns with the one used by PDBe and encompasses the following categories: (i) 'covalent;' (ii) 'electrostatic,' which combines 'ionic,' 'hbond,' 'weak_hbond,' 'polar,' 'weak_polar,' 'xbond' and 'carbonyl;' (iii) 'amide,' consisting of 'AMIDEAMIDE' and 'AMIDERING;' (iv) 'vdw,' denoting van der Waals interactions; (v) 'hydrophobic;' (vi) 'aromatic,' grouping 'aromatic,' 'FF,' 'OF,' 'EE,' 'FT,' 'OT,' 'ET,' 'FE,' 'OE,' and 'EF' contacts; (vii) 'atom-pi,' comprised of 'CARBONPI,' 'CATIONPI,' 'DONORPI,' 'HALOGENPI,' and 'METSULPHURPI;' (viii) 'metal' and (ix) 'clashes,' including 'clash' and 'vdw_clash' contacts. We have omitted this last category due to the limited number of interactions, most of which result from experimental errors during X-ray diffraction.

Backbone and side chain interactions were identified based on the atom identities in the coenzyme binding sites. Those atoms corresponding to the backbone of standard amino acids were identified as: 'N,' 'C,' 'CA,' 'O.' Glycine has only a hydrogen atom as its side chain; nevertheless, no side chain atom mediating any interaction was identified.

## Secondary structure analysis

Statistical analysis of secondary structure content was conducted at the UniProt level. For each residue within every UniProt entry, we considered all potential secondary structure elements derived from the PDB structures associated with each UniProt code. Subsequently, we eliminated redundancy on a per-residue basis. This methodological approach enabled us to comprehensively encompass the structural diversity at each position of the protein.

## Interactions mediated exclusively by early or late amino acids

To examine proteins that interacted with cofactors solely through early or late amino acids, we filtered the data to include only proteins interacting with at least two amino acids.

For the assessment of the evolutionary conservation of coenzyme-binding amino acids, we employed ConSurf (*Ashkenazy et al., 2010*; *Ashkenazy et al., 2016*). Specifically, we analyzed the msa_positional_aa_frequency files generated for each PDB structure.

## Acknowledgements

This work was supported by the Human Frontier Science Program grant HFSP-RGEC27/2023 and was carried out with the support of ELIXIR CZ Research Infrastructure (ID LM2023055, MEYS CR). ACSR and MM acknowledge support by the project 'Grant Schemes at CU' (reg. no. CZ.02.2.69/0.0/0.0/19_073/0016935), project no. START/SCI/148. Finally, we would like to thank Prof. Stephen Freeland and Prof. Janet Thornton for helpful discussions on this manuscript.

## Additional information

### Funding

| Funder | Grant reference number | Author |
| --- | --- | --- |
| MEYS CR | LM20230 | Marian Novotný |
| Human Frontier Science Program | 10.52044/HFSP.RGEC272023.pc.gr.168579 | Klára Hlouchová |
| Grant Schemes at CU | CZ.02.2.69/0.0/0.0/19_073/0016935 | Alma Carolina Sanchez Rocha<br>Mikhail Makarov |

The funders had no role in study design, data collection and interpretation, or the decision to submit the work for publication.

### Author contributions

Alma Carolina Sanchez Rocha, Conceptualization, Data curation, Formal analysis, Investigation, Visualization, Methodology, Writing – original draft, Writing – review and editing; Mikhail Makarov, Formal analysis, Visualization; Lukáš Pravda, Supervision, Methodology, Writing – review and editing; Marian Novotný, Conceptualization, Formal analysis, Supervision, Validation, Methodology, Writing – original draft, Writing – review and editing; Klára Hlouchová, Conceptualization, Resources, Formal analysis, Supervision, Writing – original draft, Project administration, Writing – review and editing

### Author ORCIDs

Alma Carolina Sanchez Rocha ⓘ https://orcid.org/0000-0001-7395-9173
Marian Novotný ⓘ https://orcid.org/0000-0001-8788-3202
Klára Hlouchová ⓘ https://orcid.org/0000-0002-5651-4874

Reviewer #1 (Public review): https://doi.org/10.7554/eLife.94174.3.sa1
Reviewer #2 (Public review): https://doi.org/10.7554/eLife.94174.3.sa2
Author response https://doi.org/10.7554/eLife.94174.3.sa3

## Additional files

### Supplementary files
Supplementary file 1. PDB codes assigned to each coenzyme class.

Supplementary file 2. Identification of coenzymes in Protein Data Bank (PDB).

Supplementary file 3. Amino Acid Composition of the Coenzyme Binding Sites. Table 2A_90 Residue Composition at 90% Sequence Identity. Table 2B_30 Amino Acid Composition at 30% Sequence Identity.

Supplementary file 4. Folds catalogue of ECOD X-groups in coenzyme binding sites.

Supplementary file 5. Proteins and nucleic acids with coenzyme binding mediated by metallic ions and water molecules.

Supplementary file 6. Amino acid fractional differences observed across all coenzyme binding sites.

Supplementary file 7. Amino acid fractional differences observed across all non-phosphate-containing coenzyme binding sites.

Supplementary file 8. Coenzymes interacting with nucleic acids.

Supplementary file 9. Chi-squared test of early versus late amino acid composition per coenzyme class.

Supplementary file 10. Chi-squared test comparing early versus late residue composition across all coenzyme temporalities in different interaction types.

Supplementary file 11. Average secondary structure content of the different coenzyme temporalities.

MDAR checklist

### Data availability
All data supporting the study are available at https://osf.io/z4svt/, including residue-level structural information for all analyzed protein-coenzyme complexes and in Supplementary Information. The code developed for the analyses presented in this paper is publicly available at https://github.com/AlmaCa-rolina-SanchezRocha/SanchezRocha-Coenzymes2024 (copy archived at *Sanchez Rocha, 2025*).

The following dataset was generated:

| Author(s) | Year | Dataset title | Dataset URL | Database and Identifier |
|---|---|---|---|---|
| Sanchez Rocha AC | 2025 | Data for 'Coenzyme-Protein Interactions Since Early Life' | https://osf.io/z4svt/ | Open Science Framework, z4svt |

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
