## [Editor Report · eLife Assessment]

This study presents a **valuable** examination of the prevalence of interactions between amino acids from different periods of Earth's history and coenzymes. While the premise of this work is well founded and the analysis is **solid**, with more data, the interpretation could change. This manuscript would be of interest to evolutionary biologists and biophysicists.

---

## [Referee Report · Reviewer #1 (Public review)]

By examining the prevalence of interactions with ancient amino acids of coenzymes in ancient versus recent folds, the authors noticed an increased interaction propensity for ancient interactions. They infer from this that coenzymes might have played an important role in prebiotic proteins. By only focusing on coenzymes, the authors may have overestimated their importance. What about other small molecules that existed in the prebiotic soup? Do they also prefer such ancient amino acids? if so, this might reflect the interaction propensity of specific amino acids rather than some possible role in very ancient proteins. Or it might diminish the conjectured importance of coenzymes. The analysis, which is very straightforward, is technically correct. However, the conclusions might not be as strong as presented. This paper presents an excellent summary of contemporary thought on what might have constituted prebiotic proteins and their properties.

---

## [Referee Report · Reviewer #2 (Public review)]

This study advances the model that the first canonical amino acids to emerge in life bound the earliest cofactors and led to the first proteins. The focus is on organic/organometallic cofactors, building on previous work on metals - ie. those in the groups of Bromberg, Dupont and others as well cited in the manuscript. Studies of this type are limited both by data availability and confounding chemical effects that are exacerbated by the timescale of evolutionary inference tackled here. However, the analysis provides a solid addition to the field and complements existing metal-focused studies as well as those Longo, Russell and others (also well cited).

---

## [Author Response]

The following is the authors’ response to the original reviews.

**Reviewer #1 (Public Review):**
Summary:By examining the prevalence of interactions with ancient amino acids of coenzymes in ancient versus recent folds, the authors noticed an increased interaction propensity for ancient interactions. They infer from this that coenzymes might have played an important role in prebiotic proteins.Strengths:(1) The analysis, which is very straightforward, is technically correct. However, the conclusions might not be as strong as presented.(2) This paper presents an excellent summary of contemporary thought on what might have constituted prebiotic proteins and their properties.(3) The paper is clearly written.

We are grateful for the kind comments of the reviewer on our manuscript. However, we would like to clarify a possible misunderstanding in the summary of our study. Specifically, analysis of "ancient versus recent folds" was not really reported in our results. Our analysis concerned "coenzyme age" rather than the "protein folds age" and was focused mainly on interaction with early vs. late amino acids in protein sequence. While structural propensities of the coenzyme binding sites were also analyzed, no distinction on the level of ancient vs. recent folds was assumed and this was only commented on in the discussion, based on previous work of others.

Weaknesses:(1) The conclusions might not be as strong as presented. First of all, while ancient amino acids interact less frequently in late with a given coenzyme, maybe this just reflects the fact that proteins that evolved later might be using residues that have a more favorable binding free energy.

We would like to point out that there was no distinction between proteins that evolved early or late in our dataset of coenzyme-binding proteins. The aim of our analysis was purely to observe trends in the age of amino acids vs. age of coenzymes. While no direct inference can be made from this about early life as all the proteins are from extant life (as highlighted in the discussion of our work), our goal was to look for intrinsic propensities of early vs. late amino acids in binding to the different coenzyme entities. Indeed, very early interactions would be smeared by the eons of evolutionary history (perhaps also towards more favourable binding free energy, as pointed out also by the reviewer). Nevertheless, significant trends have been recorded across the PDB dataset, pointing to different propensities and mechanistic properties of the binding events. Rather than to a specific evolutionary past, our data therefore point to a “capacity” of the early amino acids to bind certain coenzymes, and we believe that this is the major (and standing) conclusion of our work, along with the properties of such interactions. In our revised version, we will carefully go through all the conclusions and make sure that this message stands out, but we are confident that the following concluding sentences copied from the abstract and the discussion of our manuscript fully comply with our data:

“These results imply the plausibility of a coenzyme-peptide functional collaboration preceding the establishment of the Central Dogma and full protein alphabet evolution”

“While no direct inferences about distant evolutionary past can be drawn from the analysis of extant proteins, the principles guiding these interactions can imply their potential prebiotic feasibility and significance.”

“This implies that late amino acids would not be necessarily needed for the sovereignty of coenzyme-peptide interplay.”

We would also like to add that proteins that evolved later might not always have higher free energy of binding. Musil et al., 2021 (https://www.ncbi.nlm.nih.gov/pmc/articles/PMC8294521/) showed in their study on the example of haloalkane dehalogenase Dha A that the ancestral sequence reconstruction is a powerful tool for designing more stable, but also more active proteins. Ancestral sequence reconstruction relies on finding ancient states of protein families to suggest mutations that will lead to more stable proteins than are currently existing proteins. Their study did not explore the ligand-protein interactions specifically but showed that ancient states often show more favorable properties than modern proteins.

(2) What about other small molecules that existed in the probiotic soup? Do they also prefer such ancient amino acids? If so, this might reflect the interaction propensity of specific amino acids rather than the inferred important role of coenzymes.

We appreciate the comment of the reviewer towards other small molecules, which we assume points mainly towards metal ions (i.e. inorganic cofactors). We completely agree with the reviewer that such interactions are of utmost importance to the origins of life. Intentionally, they were not part of our study, as these have already been studied previously by others (e.g. Bromberg et al., 2022; and reviewed in Frenkel-Pinter et al., 2020) and also us (Fried et al., 2022). For example, it is noteworthy that prebiotically relevant metal binding sites (e.g. of Mg2+) exhibit enrichment in early amino acids such as Asp and Glu while more recent metal (e.g. Cu and Zn) site in the late amino acids His and Cys (Fried et al., 2022). At the same time, comparable analyses of amino acid - coenzyme trends were not available.

Nevertheless, involvement of metal ions in the coenzyme binding sites was also studied here and pointed to their bigger involvement with the Ancient coenzymes. In the revised version of the manuscript, we will be happy to enlarge the discussion of the studies concerning inorganic cofactors.

The following sentence was added in the discussion of the revised manuscript:

“This would also be true for direct interaction of early peptides/proteins and metal ions, independent of organic cofactor involvement, as discussed previously by us and others (Bromberg et al., 2022; Frenkel-Pinter et al., 2020; Fried et al., 2022). For example, it has been observed that coordination of prebiotically most relevant metal ions (e.g., Mg2+) is more often mediated by early amino acids such as Asp and Glu, whereas metal ions of later relevance (e.g., Cu and Zn) bind more frequently via late amino acids like His and Cys (Fried et al. 2022). Similarly, ancient metal binding folds have been shown to be enriched in early amino acids (Bromberg et al., 2022).”

(3) Perhaps the conclusions just reflect the types of active sites that evolved first and nothing more.

We partly agree on this point with the reviewer but not on the fact why it is listed as the weakness of our study and on the “nothing more” notion. Understanding what the properties of the earliest binding sites is key to merging the gap between prebiotic chemistry and biochemistry. The potential of peptides preceding ribosomal synthesis (and the full alphabet evolution) along with prebiotically plausible coenzymes addresses exactly this gap, which is currently not understood.

**Reviewer #2 (Public Review):**
I enjoyed reading this paper and appreciate the careful analysis performed by the investigators examining whether 'ancient' cofactors are preferentially bound by the first-available amino acids, and whether later 'LUCA' cofactors are bound by the late-arriving amino acids. I've always found this question fascinating as there is a contradiction in inorganic metal-protein complexes (not what is focused on here). Metal coordination of Fe, Ni heavily relies on softer ligands like His and Cys - which are by most models latecomer amino acids. There are no traces of thiols or imidazoles in meteorites - although work by Dvorkin has indicated that could very well be due to acid degradation during extraction. Chris Dupont (PNAS 2005) showed that metal speciation in the early earth (such as proposed by Anbar and prior RJP Williams) matched the purported order of fold emergence.As such, cofactor-protein interactions as a driving force for evolution has always made sense to me and I admittedly read this paper biased in its favor. But to make sure, I started to play around with the data that the authors kindly and importantly shared in the supplementary files. Here's what I found:Point 1: The correlation between abundance of amino acids and protein age is dominated by glycine.There is a small, but visible difference in old vs new amino acid fractional abundance between Ancient and LUCA proteins (Figure 3, Supplementary Table 3). However, the bias is not evenly distributed among the amino acids - which Figure 4A shows but is hard to digest as presented. So instead I used the spreadsheet in Supplement 3 to calculate the fractional difference FDaa = F(old aa)-F(new aa). As expected from Figure 3, the mean FD for Ancient is greater than the mean FD for LUCA. But when you look at the same table for each amino acid FDcofactor = F(ancient cofactor) - F(LUCA cofactor), you now see that the bias is not evenly distributed between older and newer amino acids at all. In fact, most of the difference can be explained by glycine (FDcofactor = 3.8) and the rest by also including tryptophan (FDcofactor = -3.8). If you remove these two amino acids from the analysis, the trend seen in Figure 3 all but disappears.Troubling - so you might argue that Gly is the oldest of the old and Trp is the newest of the new so the argument still stands. Unfortunately, Gly is a lot of things - flexible, small, polar - so what is the real correlation, age, or chemistry? This leads to point 2.

We truly acknowledge the effort that the reviewer made in the revision of the data and for the thoughtful, deeper analysis. We agree that this deserves further discussion of our data.

As invited by the reviewer, we indeed repeated the analysis on the whole dataset. First, we would like to point out that the reviewer was most probably referring to the Supplementary Fig. 2 (and not 3, which concerns protein folds). While the difference between Ancient and LUCA coenzyme binding is indeed most pronounced for Gly and Trp, we failed to confirm that the trend disappears if those two amino acids are removed from the analysis (additional FDcofactors of 3.2 and -3.2 are observed for the early and late amino acids, resp.), as seen in Table I below. The main additional contributors to this effect are Asp (FD of 2.1) and Ser (FD of 1.8) from the early amino acids and Arg (FD of -2.6) and Cys (FD of -1.7) of the late amino acids. Hence, while we agree with the reviewer that Gly and Trp (the oldest and the youngest) contribute to this effect the most, we disagree that the trend reduces to these two amino acids.

In addition, the most recent coenzyme temporality (the Post-LUCA) was neglected in the reviewer’s analysis. The difference between F (old) and F (new) is even more pronounced in Post-LUCA than in LUCA, vs. Ancient (Supplementary table 5A) and depends much less on Trp. Meanwhile, Asp, Ser, Leu, Phe, and Arg dominate the observed phenomenon (Supplementary table 5b). This further supports our lack of agreement with the reviewer’s point. Nevertheless, we remain grateful for this discussion and we will happily include this additional analysis in the Supplementary Material of our revised manuscript.

The following text (and the additional data) was included in the revised manuscript version:

“To explore the contribution of individual amino acids to this effect, fractional difference (FD) for early vs. late amino acids among the Ancient, LUCA, and Post-LUCA coenzyme binding was calculated (Supplementary Table 5). The mean FD revealed a similar trend to the amino acid composition analysis (Fig. 3). The amino acids most enriched in LUCA vs. Post-LUCA are Gly, Ser, and Leu (FD of 4.4, 4.3, and 4.1 respectively), while the most depleted include Phe, Arg, and His (FD of -11, -4.2, and -3.2) (Supplementary Table 5B).”

Point 2 - The correlation is dominated by phosphate.In the ancient cofactor list, all but 4 comprise at least one phosphate (SAM, tetrahydrofolic acid, biopterin, and heme). Except for SAM, the rest have very low Gly abundance. The overall high Gly abundance in the ancient enzymes is due to the chemical property of glycine that can occupy the right-hand side of the Ramachandran plot. This allows it to make the alternating alphaleft-alpharight conformation of the P-loop forming Milner-White's anionic nest. If you remove phosphate binding folds from the analysis the trend in Figure 3 vanishes.Likewise, Trp is an important functional residue for binding quinones and tuning its redox potential. The LUCA cofactor set is dominated by quinone and derivatives, which likely drives up the new amino acid score for this class of cofactors.

Once again, we are thankful to the reviewer for raising this point. The role of Gly in the anionic nests proposed by Milner-White and Russel, as well as the Trp role in quinone binding are important points that we would be happy to highlight more in the discussion of the revised manuscript.

Nevertheless, we disagree that the trends reduce only to the phosphate-containing coenzymes and importantly, that “the trend in Figure 3 vanishes” upon their removal. Supplementary table 6A and 6B show the data for coenzymes excluding those with phosphate moiety and the trend in Fig. 3 remains, albeit less pronounced.

The following text was included in the revised manuscript version:

“Moreover, we investigated whether the observed trend in amino acid occurrence at the binding sites was dominated by the presence of phosphate groups, which are common in many ancient cofactors except for SAM, Tetrahydrofolic acid, Biopterin, and Heme. An additional analysis therefore excluded all phosphate-containing coenzymes indicating that while the trend is less pronounced, it remains even in the absence of phosphate groups (Supplementary Table 6).”

In summary, while I still believe the premise that cofactors drove the shape of peptides and the folds that came from them - and that Rossmann folds are ancient phosphate-binding proteins, this analysis does not really bring anything new to these ideas that have already been stated by Tawfik/Longo, Milner-White/Russell, and many others.I did this analysis ad hoc on a slice of the data the authors provided and could easily have missed something and I encourage the authors to check my work. If it holds up it should be noted that negative results can often be as informative as strong positive ones. I think the signal here is too weak to see in the noise using the current approach.

We are grateful to the reviewer for encouraging further look at our data. While we hope that the analysis on the whole dataset (listed in Tables I - IV) will change the reviewer’s standpoint on our work, we would still like to comment on the questioned novelty of our results. In fact, the extraordinary works by Tawfik/Longo and Milner-While/Russel (which were cited in our manuscript multiple times) presented one of the motivations for this study. We take the opportunity to copy the part of our discussion that specifically highlights the relevance of their studies, and points out the contribution of our work with respect to theirs.

“While all the coenzymes bind preferentially to protein residue sidechains, more backbone interactions appear in the ancient coenzyme class when compared to others. This supports an earlier hypothesis that functions of the earliest peptides (possibly of variable compositions and lengths) would be performed with the assistance of the main chain atoms rather than their sidechains (Milner-White and Russel 2011). Longo et al., recently analyzed binding sites of different phosphate-containing ligands which were arguably of high relevance during earliest stages of life, connecting all of today’s core metabolism (Longo et al., 2020 (b)). They observed that unlike the evolutionary younger binding motifs (which rely on sidechain binding), the most ancient lineages indeed bind to phosphate moieties predominantly via the protein backbone.

Our analysis assigns this phenomenon primarily to interactions via early amino acids that (as mentioned above) are generally enriched in the binding interface of the ancient coenzymes. This implies that late amino acids would not be necessarily needed for the sovereignty of coenzyme-peptide interplay.”

Unlike any other previous work, our study involves all the major coenzymes (not just the phosphate-containing ones) and is based on their evolutionary age, as well as age of amino acids. It is the first PDB-wide systematic evolutionary analysis of coenzyme-amino acid binding. Besides confirming some earlier theoretical assertions (such as role of backbone interactions in early peptide-coenzyme evolution) and observations (such as occurrence of the ancient phosphate-containing coenzymes in the oldest protein folds), it uncovers substantial novel knowledge. For example, (i) enrichment of early amino acids in the binding of ancient coenzymes, vs. enrichment of late amino acids in the binding of LUCA and Post-LUCA coenzymes, (ii) the trends in secondary structure content of the binding sites of coenzyme of different temporalities, (iii) increased involvement of metal ions in the ancient coenzyme binding events, and (iv) the capacity of only early amino acids to bind ancient coenzymes. In our humble opinion, all of these points bring important contributions in the peptide-coenzyme knowledge gap which has been discussed in a number of previous studies.

**Recommendations for the authors:**
(1) By only focusing on coenzymes, the authors may have overestimated their importance. What about other small molecules that existed in the prebiotic soup? Do they also prefer such ancient amino acids? If so, this might reflect the interaction propensity of specific amino acids rather than some possible role in very ancient proteins. Or it might diminish the conjectured importance of coenzymes.

The following sentence was added in the discussion of the revised manuscript:

“This would also be true for direct interaction of early peptides/proteins and metal ions, independent of organic cofactor involvement, as discussed previously by us and others (Bromberg et al., 2022; Frenkel-Pinter et al., 2020; Fried et al., 2022). For example, it has been observed that coordination of prebiotically most relevant metal ions (e.g., Mg2+) is more often mediated by early amino acids such as Asp and Glu, whereas metal ions of later relevance (e.g., Cu and Zn) bind more frequently via late amino acids like His and Cys (Fried et al. 2022). Similarly, ancient metal binding folds have been shown to be enriched in early amino acids (Bromberg et al., 2022).”

(2) The authors should analyze whether the interactions are with similar types of amino acids in ancient versus early proteins.

While we appreciate the interesting suggestion, we would like to clarify that we did not aim to elucidate the differences between early and late protein folds - we agree that this might add an interesting perspective to our work, but we feel that it is well beyond the scope of our current study.

(3) The authors might also wish to do sequence alignments to the structures in early versus late evolving proteins to see how general this pattern of residue usage is beyond the limited set of proteins found in the PDB.

This is an interesting suggestion but similar to the previous recommendation, it is not within the scope of this study where no distinction between early and late evolving proteins has been made.

There has been a number of attempts to classify the folds as shared among Bacteria, Archea and Eukaryota or specific to one or two of these groups of organisms (https://link.springer.com/article/10.1007/s00239-023-10136-x, https://www.ncbi.nlm.nih.gov/pmc/articles/PMC9541633/) - this does not however compare easily with our time scales - where ancient ligands occur well before the last common ancestor.

We also agree the set of sequences present in the PDB is biased, but perhaps it is less biased than we have thought. The recent fantastic work https://www.biorxiv.org/content/10.1101/2024.03.18.585509v2 from Nicola Bordin and his colleagues from Orengo group attempted to classify over 200 milion structures in Alphafold database in so called Encyclopedia of Domains and they found out that nearly 80% of detected domains can be assigned to already known superfamilies in CATH (https://www.biorxiv.org/content/10.1101/2024.03.18.585509v2).

(4) The authors might wish to consider the results in Skolnick, H. Zhou, and M. Gao. On the possible origin of protein homochirality, structure, and biochemical function. PNAS 2019: 116(52): 26571-26579.

Based on the editorial recommendation, the following sentence was added in the discussion:

“It has been implied by computer simulations that coenzymes could bind to proteins with similar propensity even before the onset of protein homochirality, despite lower structural stability and secondary structure content in heterochiral polypeptides (Skolnick et al., 2019).”